# Investigating the Effects of Super Typhoon HAGIBIS in the Northwest Pacific Ocean Using Multiple Observational Data

**Jonghyeok Jeon [1] and Takashi Tomita [2],***

1  Graduate School of Environmental Studies, Nagoya University, Nagoya 464-8601, Japan
2  Disaster Mitigation Research Center, Nagoya University, Nagoya 464-8601, Japan
*  Correspondence: tomita.takashi@nagoya-u.jp; Tel.: +81-52-789-2072

**Abstract:** Various multi-source observational platforms have enabled the exploration of ocean dynamics in the Northwest Pacific Ocean (NPO). This study investigated daily oceanic variables in response to the combined effect of the 2019 super typhoon HAGIBIS and the Kuroshio current meander (KCM), which has caused economic, ecological, and climatic changes in the NPO since August 2017. During HAGIBIS, the six-hourly wind speed data estimated a wind stress power ($P_w$) which strengthened around the right and left semicircles of the typhoon, and an Ekman pumping velocity (EPV) which intensified at the center of the typhoon track. As a result, firstly, the sea temperature (ST) decreased along a boundary with a high EPV and a strong cyclonic eddy area, and the mixed layer depth (MLD) was shallow. Secondly, a low sea salinity (SS) concentration showed another area where heavy rain fell on the left side of the typhoon track. Phytoplankton bloom (PB) occurred with a large concentration of chlorophyll a (0.641 mg/m$^3$) over a wide extent (56,615 km$^2$; above 0.5 mg/m$^3$) after one day of HAGIBIS. An analysis of a favorable environment of the PB's growth determined the cause of the PB, and a shift of the subsurface chlorophyll maximum layer (SCML; above 0.7 mg/m$^3$) was estimated by comprehensive impact analysis. This study may contribute to understanding different individually-estimated physical and biological mechanisms and predicting the recurrence of ocean anomalies.

**Keywords:** Northwest Pacific Ocean; Kuroshio current meander; mesoscale cyclonic eddy; typhoon HAGIBIS; wind stress power; Ekman pumping velocity; sea temperature; sea salinity; phytoplankton bloom





## 1. Introduction

The Northwest Pacific Ocean (NPO) has the highest frequency of tropical cyclonic activity worldwide. The region experiences an annual average of 16.5 events, of which 6.3 (38%) are super typhoons [1], defined as category five storms with maximum sustained winds of 150 mph (241 km/h), a minimum central pressure of 910 hPa, and accompanied by heavy rainfall and storm surges. Upper oceanic responses to super typhoons have been a critical issue due to their importance in climate change, environmental variability, and preservation of marine resources [2]. Many studies have shown that strong winds induce changes in oceanic variables such as temperature, salinity, and nutrients beneath storms through the vertical entrainment process. The energy transferred from the atmosphere to the ocean induces vertical stirring and upwelling activities [3–5]. Vertical mixing largely explains the decrease in surface temperature in the open ocean, while the Ekman upwelling plays an important role in the sea subsurface [6]. The role of super typhoons in the NPO has received increased attention across various subdisciplines of ocean physics.

At the sea surface in the Northern Hemisphere, decrease in sea temperature (ST) [7], high concentration of sea salinity (SS) [8], deepening of mixed layer depth (MLD) [9], and phytoplankton blooms (PB) estimated by a chlorophyll a (Chl-a) concentration [10] are generally more observable on the right side of the track of typhoons. Typhoon-induced

heavy rainfall reduces the amount of SS on the left side of the path of typhoons in the upper ocean [11]. At the sea subsurface, warm ST anomalies caused by the influence of the typhoon can reach approximately 4 °C [12], and the SS decreases within 1 psu [13]. Regarding biology, some studies have revealed that the subsurface chlorophyll maximum layers (SCML) contribute substantially to the growth of upper oceanic phytoplankton biomass under appropriate conditions comprising light intensity, nutrient-flux, and primary production [14]. The SCML's location, depth, and duration are modulated by vertical mixing, advection, and upwelling [15,16]. The emergence of physical and biological changes in response to typhoons is well established; however, the cause of the upper oceanic anomalies that depend on typhoon effects and ocean environmental conditions such as heat transfer current, mesoscale eddies, and geographical features have remained unclear. Influences from energy-aggregated systems such as typhoons and the marine environment and their interpretation are still lacking in comprehensive and quantitative terms.

One of the background conditions of the NPO is the large Kuroshio current meander (KCM) that transits from low latitudes to offshore southern Japan, influencing ST, SS, water density, and mesoscale eddies. Since October 2017, the large KCM has flowed southward off Shikoku Island through the Tokara Strait, and then turned back northward to the Kanto region [17]. The KCM has been going on for four years and ten months as of June 2022 [18]. This abnormal trajectory has made the south of Japan a hazardous and often unpredictable environment for the local marine ecosystem and fishing grounds [19]. Further, the KCM is emerging as a cause of climatic impacts in different regions within Japan, leading to locally varying effects such as a warm sea temperature and high air humidity in the Kanto region [20]. However, the influences of typhoons in combination with the KCM remain largely unknown in the NPO.

Upper oceanic responses to typhoons and marine conditions are becoming better understood due to various observational methods and technological advancements. Specifically, the use of observation platforms such as buoys and moorings since the 1940s [21], satellite remote sensing since the 1980s [22], and biogeochemical Argo (BGC-Argo) floats since the 2000s [23] have been used to investigate the interaction between typhoons and regional seas regarding physical and biogeochemical marine factors. However, some weaknesses in ocean technology exist such as missing values due to cloud coverage [22], biased values due to individual satellite data algorithms [24], and temporal gaps such as weekly, 5-day, and 3-day mean datasets [25]. Currently, the Copernicus Marine Environment Monitoring Service (CMEMS) can provide 6-hourly wind speed and daily ocean variables data via a combination of multi-observational platforms and oceanic/atmospheric models [26].

To better understand the relationship of various upper marine physical and biological variables having distinct environmental characteristics in response to typhoon's effects, this study firstly investigated the cause analysis of physical and biological processes through surface ocean changes in ST, SS, MLD, and PB in response to the 2019 super typhoon HAGIBIS under KCM conditions. Secondly, it determined supplementary biological mechanisms for favorable PB growth conditions and synthesized the response of sea subsurface ocean variables during the typhoon. The following research questions are discussed in this study;

- Is it possible to interpret the changes in daily ocean variables in response to typhoon HAGIBIS under the KCM?
- Which typhoon factor is responsible for physical and biological ocean responses such as ST, SS, MLD, and PB on the sea surface?
- Is it possible to distinguish the favorable environment of the PB and interpret the cause of the widespread occurrence of a PB using a comprehensive impact analysis before and after a typhoon?

The remainder of the manuscript is organized as follows. Section 2 briefly describes the information of the study area, HAGIBIS, and multi-source datasets, and outlines the adopted methodology. Section 3 presents the results of the impact of the typhoon and the KCM on the individually-estimated sea surface and subsurface oceanic variables responses. Then, it shows the appearance mechanisms of physical and biological factors using a

comprehensive impact analysis. Finally, Section 4 highlights the key results obtained, and the implications.

## 2. Materials and Methods

### 2.1. Target Event and Region

Super Typhoon HAGIBIS and the Study Area

The best track of HAGIBIS is depicted in Figure 1, obtained by the Japan Meteorological Agency (JMA; https://www.data.jma.go.jp/fcd/yoho/typhoon/index.html accessed on 25 March 2022). The estimated typhoon factors include the following information in Figure 1a; the location of the center of the typhoon, date, central pressure/maximum sustained wind speed (m/s), and the radius of the storm wind zone (SWZ), which is defined as a wind speed of 25 m/s or more. HAGIBIS was the strongest tropical cyclone in the Northwest Pacific in 2019, and was classified as a class 5 storm with a maximum sustained wind of 160 mph (260 km/h) in the open ocean.

The coordinates for the study area are longitude 134°–143°E and latitude 30°–37.5°N, with a boundary including the central HAGIBIS track and the KCM as shown in Figure 1b. A daily KCM derived by the Japan Coast Guard (JCC, red dotted line; https://www1.kaiho.mlit.go.jp/KANKYO/KAIYO/qboc/ accessed on 1 April 2022) is described as the environmental feature in the offshore South Japan archipelago during the typhoon period. The maximum sustained wind speed was reduced from under 45 to 35 m/s, central pressure was weakened from 945 to 965 hPa passing through south of Japan, and another important typhoon factor was the translation speed (TS; km/h), which increased when HAGIBIS approached the Japanese archipelago (from 16.7 to 58.2 km/h) with a wide SWZ (radius of storm wind; from 324 to 296 km). Regarding duration, HAGIBIS constantly affected the study region for 30 h.

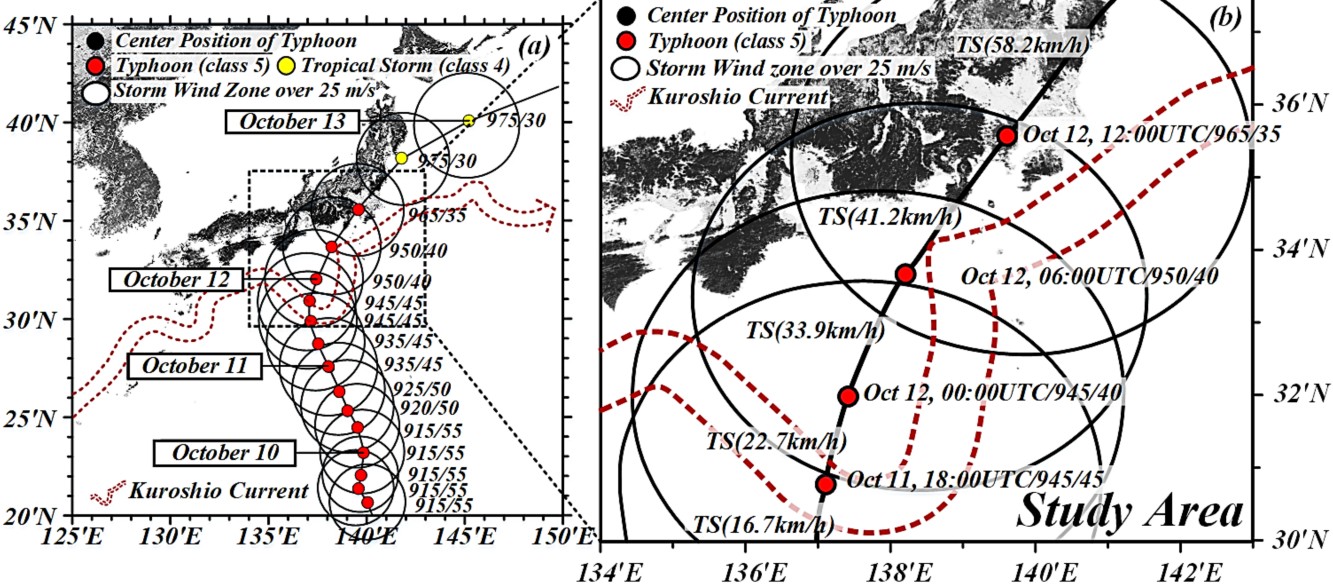

**Figure 1.** Six-hourly HAGIBIS best track on 9–13 October 2019 obtained by JMA and daily Kuroshio current meander (KCM; red dotted line) provided by JCC (**a**) at the wide area; including the typhoon central location with intensity (dots in red to yellow), the radius of the storm wind zone (SWZ; black circles), date (month/day), and central pressure (hPa)/maximum sustained wind speed (m/s), and (**b**) at the boundary of the study area; including same components as (**a**) and adding time (UTC) and translational speed (TS; km/h), respectively.

*2.2. Multi-Source Observational and Model Data*

2.2.1. Ocean Wind Product Data and Estimated Typhoon Effects

To evaluate the effect of HAGIBIS on the open ocean, we calculated the wind stress power ($P_w$), which causes the breaking of stratified layers forced by vertical wind stirring in an upper ocean, and an Ekman pumping velocity (EPV; $\times 10^{-5}$ m/s), which can drive the upwelling of temperature, water density, and nutrients from a subsurface layer to an upper ocean layer. The two factors were estimated in terms of wind speed and wind stress data.

- Wind speed and wind stress data

The global blended mean CMEMS wind product comprised six hourly averaged fields for sea surface above 10 m wind speeds, wind zonal/meridional components, wind stress zonal/meridional components, and wind stress curl. Firstly, the near-real-time (NRT) L4 product for sea surface wind factors was derived from scattermeters on board ASCAT-A and ASCAT-b for coastal winds. Secondly, remote wind speeds were gathered by the SSMIS radiometer on board the F16, F17, F18, and F19 satellites. Finally, wind speed and direction data were collected by a WindSat radiometer on board the Coriolis satellite. The blended dataset was performed for each synoptic time (00UTC, 06UTC, 12UTC, and 18UTC) with a spatial resolution of 0.25° in longitude and latitude over the global ocean (https://marine.copernicus.eu/ accessed on 8 June 2022).

- Equations

The $P_w$ in the upper ocean layer is a result of the wind-generated kinetic energy flux (W/m$^2$) used by Pan et al. [27];

$$P_w = \tau_0 U_{10} = \rho_{air} C_D U_{10}^3 \tag{1}$$

where $\tau_0$ is the wind stress at the sea surface ($\tau_0 = \rho_{air} C_D U_{10}^2$); $U_{10}$ is the wind speed at 10 m height on the mean sea level provided by the CMEMS dataset; $\rho_{air}$ is the density of air ($\approx 1.22$ kg/m$^3$ at 20 °C); and $C_D$ is the drag coefficient.

The EPV is calculated in terms of the quantitative wind stress curl, defined in (Price et al. [3]);

$$EPV = Curl(\frac{\tau_0}{f \times \rho_w}) \tag{2}$$

where $f$ is the Coriolis parameter ($f = 2\Omega \sin \phi$); $\Omega$ is Earth's rotation vector ($2\pi$ radians per sidereal day); $\phi$ is the latitude; $\rho_w$ is the density of seawater (1025 kg/m$^3$).

2.2.2. Surface Ocean Variables

The composite observational and model data were sufficient to be able to monitor the influence of typhoons and to describe each response of the upper ocean variables. The surface oceanic and meteorological data are separated into three parts: (1) Model data; sea surface temperature (SST), sea surface salinity (SSS), MLD, and Chl-a. (2) Satellite data; sea level anomaly (SLA) with geostrophic current velocity (GV) represents the KCM. (3) Radar data; daily cumulative rainfall.

- SST, SSS, and MLD

The quality of GLOBAL_ANALYSISFORECAST_PHY_CPL_001_015, which is the global physical analysis and coupled forecasting product provided by the CMEMS, is composed of 3D daily mean fields of temperature and salinity, zonal and meridional velocities, as well as 2D daily mean fields of sea surface height, bottom temperature, and MLD since 30 December 2015. The daily forecasts are produced using a coupled atmosphere-ocean system, resulting in a mean interpolated to a regular 1/4-degree latitude-longitude grid and 43 vertical levels extracted by sea water potential temperature and seawater salinity. The depth of the mixed layer is generally calculated using a hydrographic method conducted by water property measurements with two parameters of temperature

and water density shift from a reference value on the surface. For this study, the density threshold as $\Delta\theta = 0.8\ ^\circ$C decrease was used as a reference temperature by Kara et al. [28].

- Chl-a

The quality of global biogeochemistry analysis and forecast product is called GLOBAAL_ANALYSIS_FORECAST_BIO_001_028. This product is supplied on a regular grid at $0.25^\circ \times 0.25^\circ$ with 50 vertical levels on the global ocean. It includes daily fields of the following biological and biogeochemical variables: concentration of nitrates, phosphates, silicates, iron, dissolved oxygen, primary production, chlorophyll, pH, and surface partial pressure of carbon dioxide. Specifically, the sea surface and vertical distributions of the modeled chlorophyll fields showed significant correlations with satellite data and BGC-Argo measurements with a correlation coefficient of 0.81 RMSD 0.59 in the global ocean (https://marine.copernicus.eu/ accessed on 4 March 2021).

- SLA and GV

To identify the KCM and mesoscale oceanic eddies, the SLA data, also known as the near-real-time-global-altimetry dataset (version NRT 3.0 exp), provided by AVISO (Arching, Validation, and Interpretation of Satellite Oceanographic) has been available since April 2019 to present. A cyclonic (anticyclonic) eddy is identified with the contours of maximum $-1.00$ ($+1.00$) m above the mean surface level, divergence (convergence), and anti-clockwise (clockwise) based on the Northern hemisphere with GV (m/s), which is used by the meridian (V) and zonal components (U) under format for inferring the magnitude and direction at the ocean surface (http://www.aviso.altimetry.fr/en/home.html accessed on 8 June 2022).

- Daily cumulative precipitation

The tropical rainfall measuring mission (TRMM) precipitation radar (PR) observes the daily accumulated precipitation with gauge calibration as mm units. This product is generated from a research-quality three hourly TRMM multi-satellite precipitation analysis (TMPA; 3B42 L3 version 7). The period is covered by one daily particle amount per 24 h, which includes a grid at $0.25^\circ \times 0.25^\circ$, provided by the Goddard Earth Sciences Data and Information Services Center (GESDISC; https://disc.gsfc.nasa.gov/datasets/TRMM_3B42_Daily_7/summary accessed on 2 May 2022).

2.2.3. Vertical Profile of Subsurface Ocean Variables

An Argo observational system is a global and local scale array of temperature, salinity, and biogeochemical profiling floats and is planned as a major component of the ocean observing system, capable of surveying the upper 2000 m. This study addressed three Argo floats, verifying physical and biological ocean surface model data, and estimating a favorable environment for the growth of a PB. The oceanic in situ data are obtained from platform numbers 2,903,367 and 2,093,376 provided by JMA's Argo and code 2,902,754 supported by the Ministry of Science and Technology (MOST)'s Argo of China. The data is extracted from the Copernicus Marine in situ TAC dataset (http://www.marineinsitu.eu accessed on 31 May 2022) and the real-time Argo database of the China Argo Real-time Data Center (http://www.argo.org.cn accessed on 23 June 2020), respectively (Table 1).

**Table 1.** Three Argo floats information used in this study.

| Platform Code | Available Date | Parameters | Source |
|---|---|---|---|
| Physical 2903367 | 28 May 2019 to 20 July 2020 | Sea pressure, temperature, and practical salinity | ftp://nrt.cmems-du.eu/Core/ INSITU_GLO_NRT_ OBSERVATIONS_013_030/glo_ multiparameter_nrt/monthly/PF/ 201910/GL_PR_PF_2903376_2019 10.nc (accessed on 31 May 2022) |
| Physical 2903376 | 3 August 2019 to 1 October 2020 | Sea pressure, temperature, and practical salinity | ftp://nrt.cmems-du.eu/Core/ INSITU_GLO_NRT_ OBSERVATIONS_013_030/glo_ multiparameter_nrt/monthly/PF/ 201910/GL_PR_PF_2903367_2019 10.nc (accessed on 31 May 2022) |
| BGC 2902754 | 30 August 2018 to 10 February 2021 | Physical (pressure, temperature, salinity), biogeochemical (DO, nitrate, and Chl-a) | http://www.ifremer.fr/co-argoFloats/float?ptfCode=2902754 (accessed on 23 June 2020) |

*2.3. Methodology*

A stepwise framework in this study is shown in Figure 2. We proceeded with the following four steps:

- [Step 1] The typhoon effects on the NPO were estimated in terms of $P_w$ and EPV. Firstly, six-hourly wind speed data was validated by JMA data (1.1). The visualized spatial distribution shows the magnitude, direction, and impacted area of strong $P_w$ and high EPV every six hours (1.2).
- [Step 2] To interpret the physical KCM features such as meander and eddies, we reproduced the KCM expressed by SLA and GV and validated the KCM by another trajectory data analyzed by JCC (2.1). Further, the spatial distribution of SLA and GV detected the area of existing cyclonic and anticyclonic eddies (2.2). Then, we estimated the vertical variability of KCM based on 0, 60, and 100 m depths during HAGIBIS.
- [Step 3] We explored the response of ocean variables near the sea surface, considering SST, SSS, MLD, and Chl-a as well as daily cumulative rainfall. This was conducted to validate the global CMEMS model data in the study area through in situ data (3.1). The changes in surface variables (SST, SSS, Chl-a) can be estimated by where, when, and to what extent the typhoon affects the study area (3.2). The following work was used through the MLD and daily precipitation to evaluate other significant evidence (3.3).
- [Step 4] To interpret the sea surface PB one day after HAGIBIS, this study was expanded using vertical profiles showing the subsurface ocean variabilities according to a specific depth. These findings were conducted largely in two parts. Firstly, it was evaluated as a favorable environmental condition for PB growth via Argo float data (4.1). Secondly, the overall estimated spatial distribution was presented in a quantitative conceptual diagram using a comprehensive impact analysis to identify the biological growth process of the PB in the upper ocean (4.2).

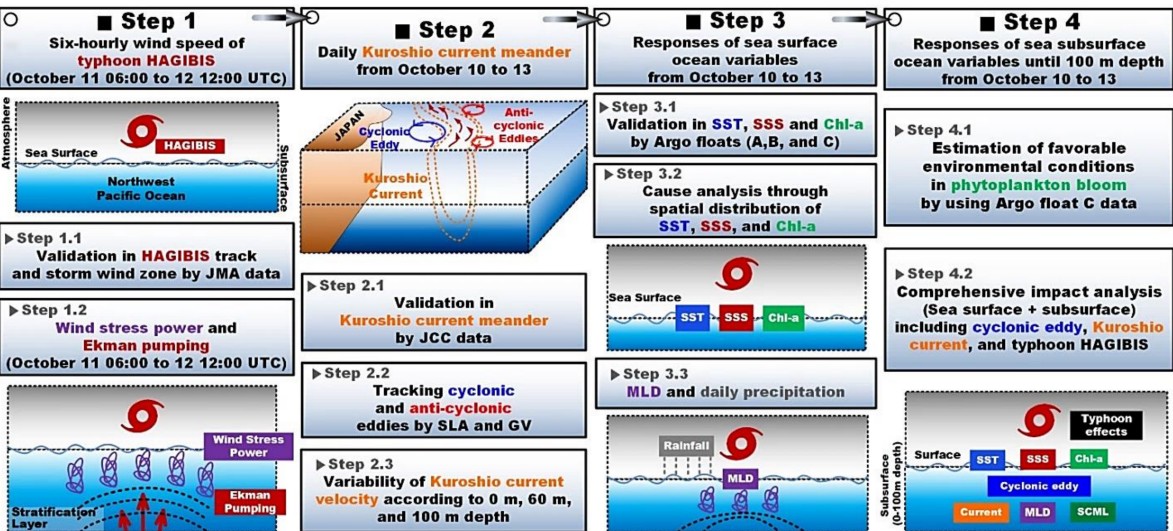

**Figure 2.** Flowchart depicting the individually-estimated physical and biological mechanisms at the environment of the Northwest Pacific Ocean in response to typhoon HAGIBIS.

## 3. Results

### 3.1. Wind Effects Induced by HAGIBIS

#### 3.1.1. Validation of CMEMS Model Data by Comparison with JMA

Global CMEMS data were compared to JMA data based on the radius of SWZs in Figure 3. The JMA provided a SWZ of above 25 m/s using a Himawari-8 geostationary satellite image [29], whereas the global ocean wind blended data obtained by the CMEMS dataset showed a white lines of 23 m/s. Comparing the two data revealed that the differences between CMEMS and JMA were approximately 2 m/s. Thais is consistent with the bias difference between the in situ and blended wind data evaluated by Bentamy et al. [30]. Furthermore, Figure 3a,b,d indicated the daily asymmetric typhoon's activities depicting the different affected areas on the typhoon's left and right semicircle. However, Figure 3c illustrates that the blended wind speed in CMEMS demonstrates that the distribution is underestimated in nearshore sites (<50 km in shoreline) because geographical interpolation is conducted close to the shoreline [30].

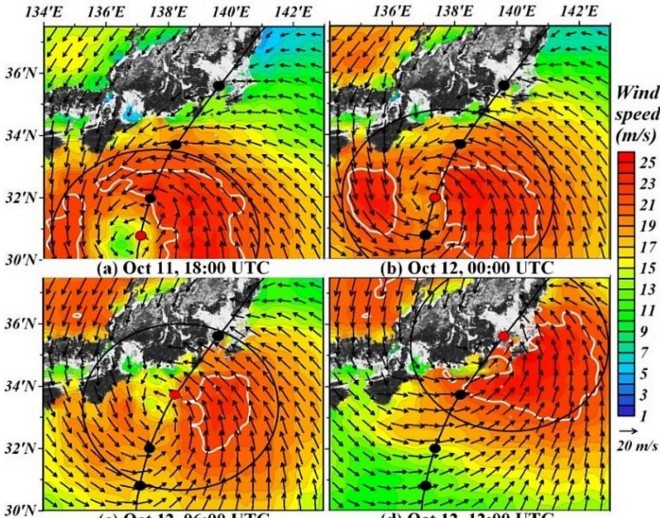

**Figure 3.** Validation of each six-hourly storm wind zone of HAGIBIS between JMA (black circles) and CMEMS data (white contours) with the typhoon central location according to time; (**a**) 11 October, 18UTC, (**b**) 12 October, 00UTC, (**c**) 06UTC, and (**d**) 12UTC, respectively.

### 3.1.2. Wind Stress Power ($P_w$) and Ekman Pumping Velocity (EPV)

To evaluate the impact of HAGIBIS on the NPO, Figure 5 illustrates the spatial distribution of $P_w$ corresponding to the HAGIBIS track. Figure 5 shows a more expanded time from 11 October, 06UTC to 12 October, 12UTC compared with Figure 3. The strong $P_w$ appears in the study area on 11 October, 06UTC. It shows that the strong $P_w$ affected the ocean along the typhoon track, except for the center of the HAGIBIS track (Figure 5b). Then, the $P_w$ intensity is sharply reduced on the typhoon's left semicircle, while another $P_w$ on the right semicircle continues to have a substantial impact (Figure 5c,d). When HAGIBIS entered the study area boundary, the translation speed was slower than when traveling long distances across the ocean (Figure 1a), and the movement going straight northward close to Japan became oblique with Japan. In this process, the $P_w$ on the left semicircle decreased faster than on the right side. When the typhoon made landfall on the Japanese archipelago, the value of $P_w$ was weakened due to the nearshore bias (Figure 5e). When exiting from Japan, the $P_w$ increased from the right side of the typhoon (Figure 5f).

Meanwhile, the physical value of EPV is shown in Figure 4: upwelling in red and downwelling in blue contour, respectively. When the HAGIBIS's effect became visible, as shown in Figure 4a, the upwelling appeared within the SWZ, while the downwelling occurred outside the SWZ. Furthermore, the strong upwelling was distributed in the study area along the center location of the typhoon track ($5 \times 10^{-5}$ m/s at maximum value, invisible in Figure 4c). The upwelling energy was concentrated at the typhoon's center. When making landfall, as shown in Figure 4e,f, the HAGIBIS still had strong upwelling along the typhoon's track. To summarize the influence of the typhoon, the $P_w$ was much stronger on the right and left semicircle than on the typhoon's center, while Ekman pumping was much stronger along the typhoon's center.

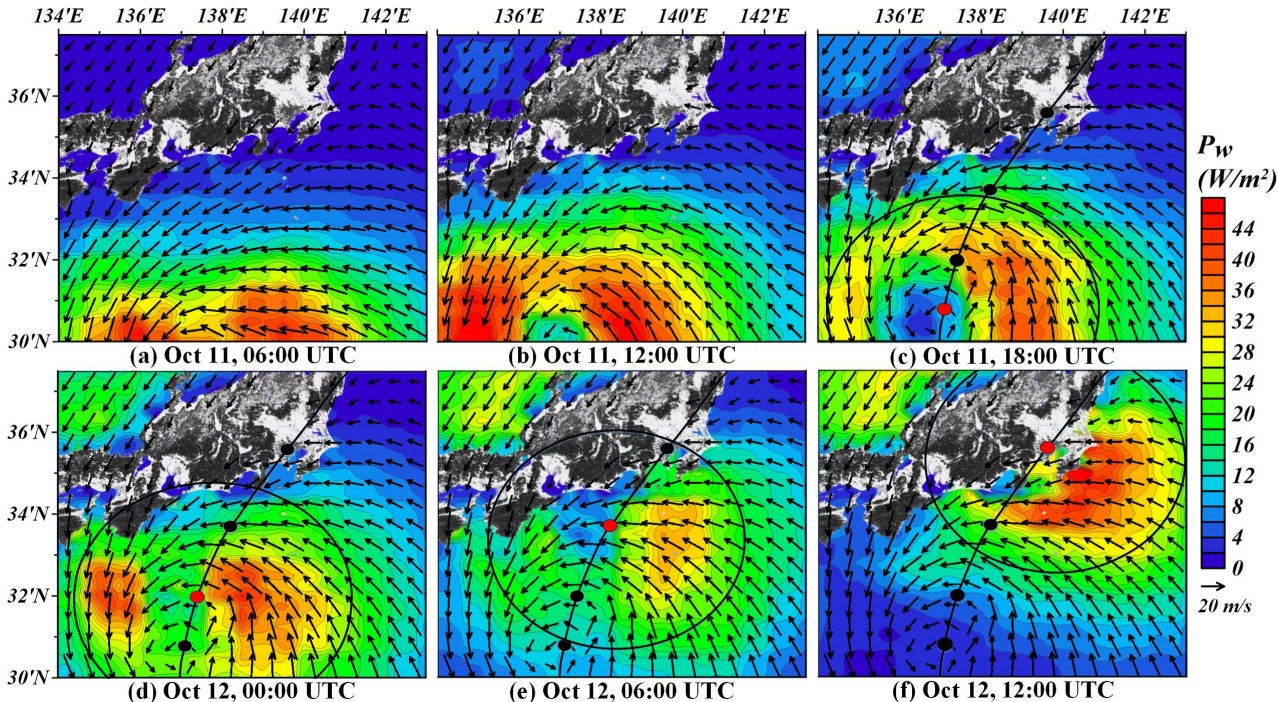

**Figure 4.** Spatial distribution of Ekman pumping velocity (EPV; $\times 10^{-5}$ m/s) with wind stress vectors (black arrows) provided by CMEMS, storm wind zones (black circles), and the typhoon track with the intensity (black line; red dots) obtained by JMA; (**a**) 11 October, 06UTC, (**b**) 12UTC (**c**) 18UTC, (**d**) 12 October, 00UTC (**e**) 06UTC, and (**f**) 12UTC, respectively.

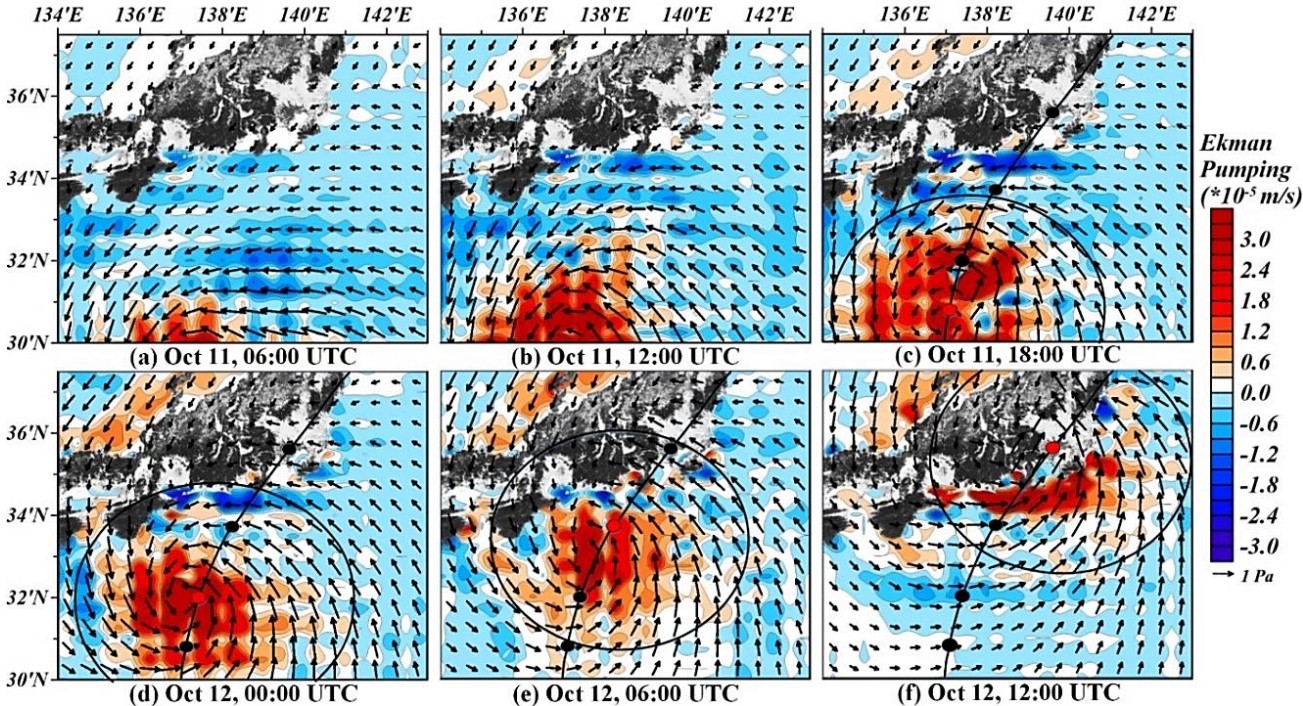

**Figure 5.** Spatial distribution of wind stress power $\left(P_w; \text{W/m}^2\right)$ with wind speed vectors (black arrows) provided by CMEMS, storm wind zones (black circles), and the typhoon track with the intensity (black line; red dots) obtained by JMA; (**a**) 11 October, 06UTC, (**b**) 12UTC (**c**) 18UTC, (**d**) 12 October, 00UTC (**e**) 06UTC, and (**f**) 12UTC, respectively.

*3.2. Environmental Condition in the NPO*

3.2.1. Validation of the KCM and Tracking Cyclonic and Anticyclonic Eddies

From existing environmental conditions ahead of HAGIBIS, this study focused on the KCM represented by the SLA and the GV. In Figure 6, we first confirmed that the quantitative Kuroshio mainstream (red arrows; above 1 m/s) provided by the AVISO dataset was well compared to JCC's estimation on 10–13 October. Before the arrival of HAGIBIS (Figure 6a), pre-existing cyclonic (counterclockwise; -value, blue shades) and anticyclonic (clockwise; +value, red shades) eddies existed along the Kuroshio's passage, influencing the surface ocean heat and generating the cyclonic and anticyclonic eddies. The intensive cyclonic eddy region with a maximum depression of −1.11 m (longitude 137.8°E and latitude 31.45°N) was mainly located inside the KCM, whereas anticyclonic eddies were along and outside the KCM (longitude 139.3°E and latitude 33.7°N with + 0.50 m, and longitude 142.7°E and latitude 35.5°N with + 0.58 m). During the typhoon, the distribution is depicted in Figure 6c compared to before the typhoon (Figure 6a). Anomalies in the intensity and the extent of eddies were not apparent. The mesoscale eddies were less affected by HAGIBIS than the KCM. Sun et al. [31] suggested that 49 super typhoons passed over 192 cyclonic eddies in the NPO from 2000 to 2008, and the typhoons intensified only approximately 10% of these eddies. This study provides additional evidence of an ineffective effect on the strength of cyclonic eddies, despite the typhoon staying at the strong cyclonic eddy area for two days.

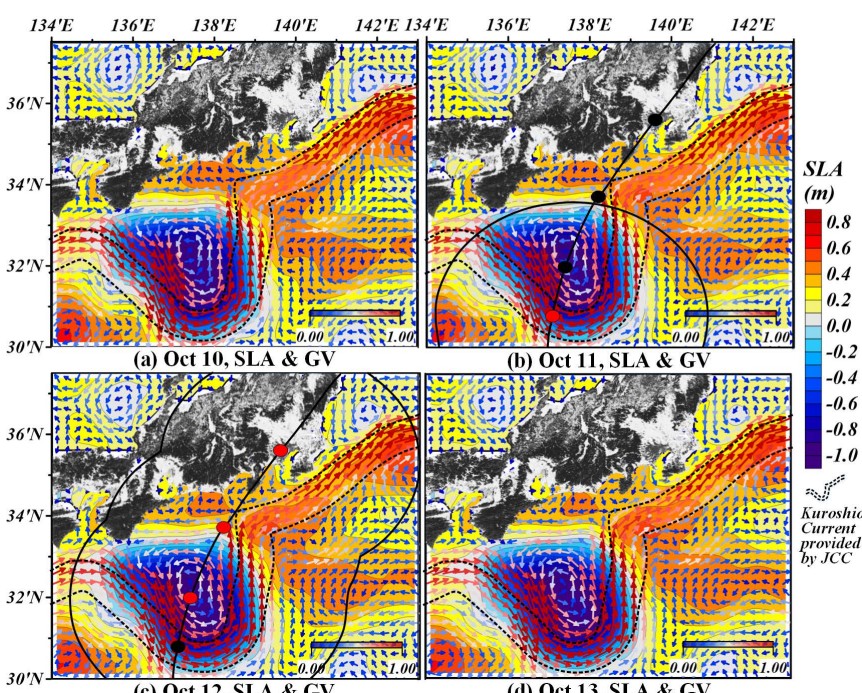

**Figure 6.** Validation of the daily Kuroshio mainstream between AVISO (red arrows) and JCC (black dotted line) data and tracking cyclonic and anticyclonic eddy in terms of SLA; (**a**) 10 October, (**b**) 11 October, (**c**) 12 October, and (**d**) 13 October with the typhoon track and intensity, respectively.

### 3.2.2. Variability of the KCM According to 0, 60, and 100 m Depth

We conducted the study of the variability of KCM at 0, 60, and 100 m depths to estimate the influences of the typhoon on the KCM (Figure 7). We focused on an area where the strong Kuroshio current flows from longitude 135°–140E° and latitude 30°–34°N. Before the typhoon (Figure 7(a1)), the KCM flowed southeastward before turning northward at a speed of over 1 m/s, with a significant horizontal distribution, reaching a depth of 100 m (Figure 7(a2,a3)). On 11 October, the strong $P_w$ on the right-hand side intensified Kuroshio's current velocity and extended the area (deeper red area; longitude 138°–139.5°E and latitude 30°–34°N). Specifically, the wind stress vector turned counterclockwise on the typhoon track's right side, intensifying current velocities in the same direction. This is caused when wind stress power greatly forces the right semicircle, and the movement's direction counterclockwise is similar to Kuroshio's path. In contrast, the direction of the typhoon's motion was opposite to Kuroshio's direction, weakening the current velocity (Figure 7(b1); on the left side of the typhoon track). The effect of $P_w$ became weaker as depth increased, and the velocity distribution at 100 m depth was no longer similar to the one before the typhoon. In addition, the high EPV on the center of the typhoon's track can induce the current velocity to increase from 100 m depth. By making landfall on Japan Island on 12 October (Figure 7(c1)), HAGIBIS drove the flow in the same direction as the Kuroshio current flow (counterclockwise) on the left side of the typhoon's track. It is assumed that the typhoon's rotation on the navigable semicircle and the current's direction coincide. Meanwhile, at the center of HAGIBIS across the Kuroshio path, the current velocities were reduced because the $P_w$ on the center of the typhoon was weaker than on the right and left semicircles of the typhoon. With an increased depth of 100 m, the KCM tended to be similar to before the typhoon. On 13 October, as shown in a column of Figure 7(d1–d3), the current velocities returned to before the typhoon while the expanded area (over 0.5 m/s; green area) was irrecoverable compared with 10 October.

Overall, at 0 and 60 m depth, the high current velocity had a larger area (above 1.0 m/s) induced by wind stirring in Figure 7(b1,c1) and then recovered at 100 m depth to before the typhoon condition (Figure 7(a3,b3,c3,d3)). The vertical fea-

ture of Kuroshio current velocities agrees with the previous result of Liu et al. [32] about the deepening depth and the distribution of the Kuroshio current.

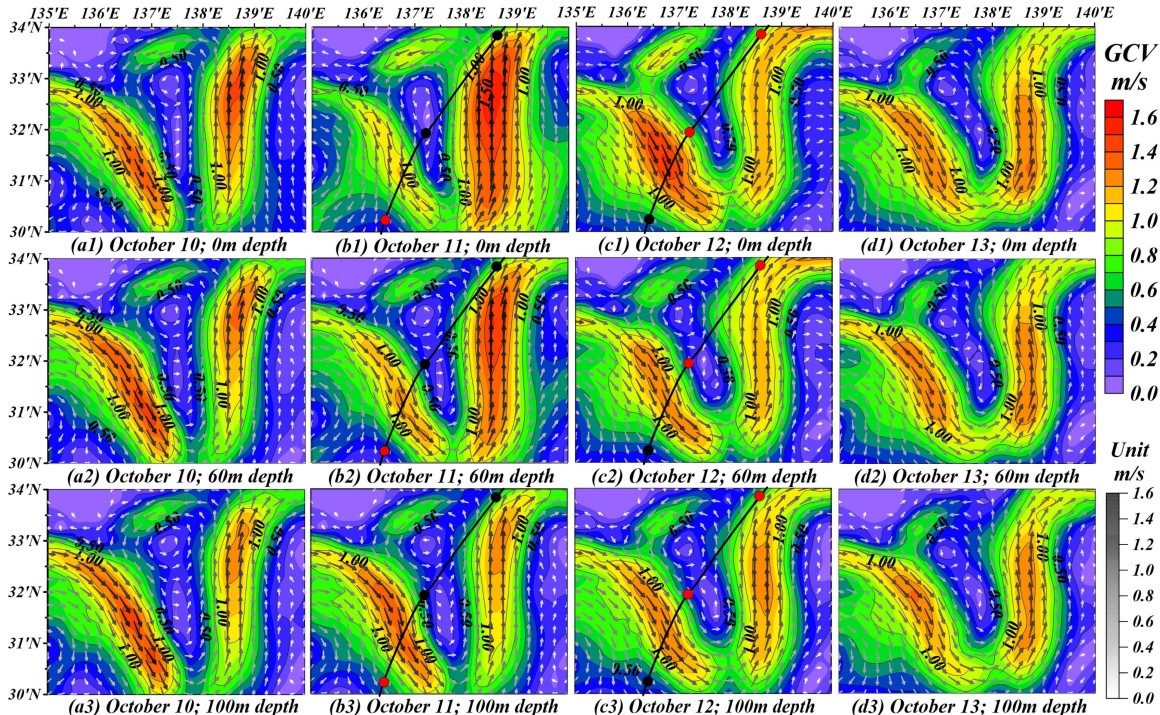

**Figure 7.** Kuroshio current velocity (m/s) on the boundary of longitude 135–140°E and latitude 30–34°N based on depth and date; 0 m (**a1,b1,c1,d1**), 60 m (**a2,b2,c2,d2**), and 100 m (**a3,b3,c3,d3**) on 10, 11, 12, and 13 October, respectively. The figures are focused on the strong cyclonic eddy area with the typhoon's track and intensity (black line and red dots).

*3.3. Responses of Sea Surface Ocean Variables*

3.3.1. Validation of SST, SSS, and Chl-a by In-Situ Data

To estimate surface ocean variables in response to the typhoon, we validated the CMEMS model data such as SST, SSS, and biological data such as Chl-a compared with in situ data to apply to the study area. Table 2 indicates the Argo floats at A, B, and C1 including locations, dates, and values. The comparative result for SST has a difference ranging from −0.11 to −0.06 °C, for SSS, a distinction ranging from −0.01 to +0.06 psu, and for Chl-a, a difference of +0.05. There is no significant difference in value between in situ and CMEMS data.

**Table 2.** Validation of oceanic CMEMS values in comparison to in-situ data.

| Factor | Argo Floats | Latitude(°N) | Longitude(°E) | Date | In-Situ Value | CMEMS Value |
|---|---|---|---|---|---|---|
| SST (°C) | A * 2903367 | 30.504 | 135.767 | 10 October | 28.00 | 27.89 (−0.11) |
| | B * 2902754 | 33.580 | 138.607 | 9 October | 26.47 | 26.41 (−0.06) |
| | C1 * 2903376 | 33.716 | 137.403 | 9 October | 26.17 | 26.06 (−0.11) |
| SSS (psu) | A 2903367 | 30.504 | 135.767 | 10 October | 34.52 | 34.46 (−0.06) |
| | B 2902754 | 33.580 | 138.607 | 9 October | 33.95 | 34.01 (+0.06) |
| | C1 2903376 | 33.716 | 137.403 | 9 October | 34.02 | 34.01 (−0.01) |
| Chl-a (mg/m³) | C1 2903376 | 33.716 | 137.403 | 9 October | 0.30 | 0.35 (+0.05) |

* The locations of Argo floats (A, B, and C1) are shown in Figure 8a1,b1.

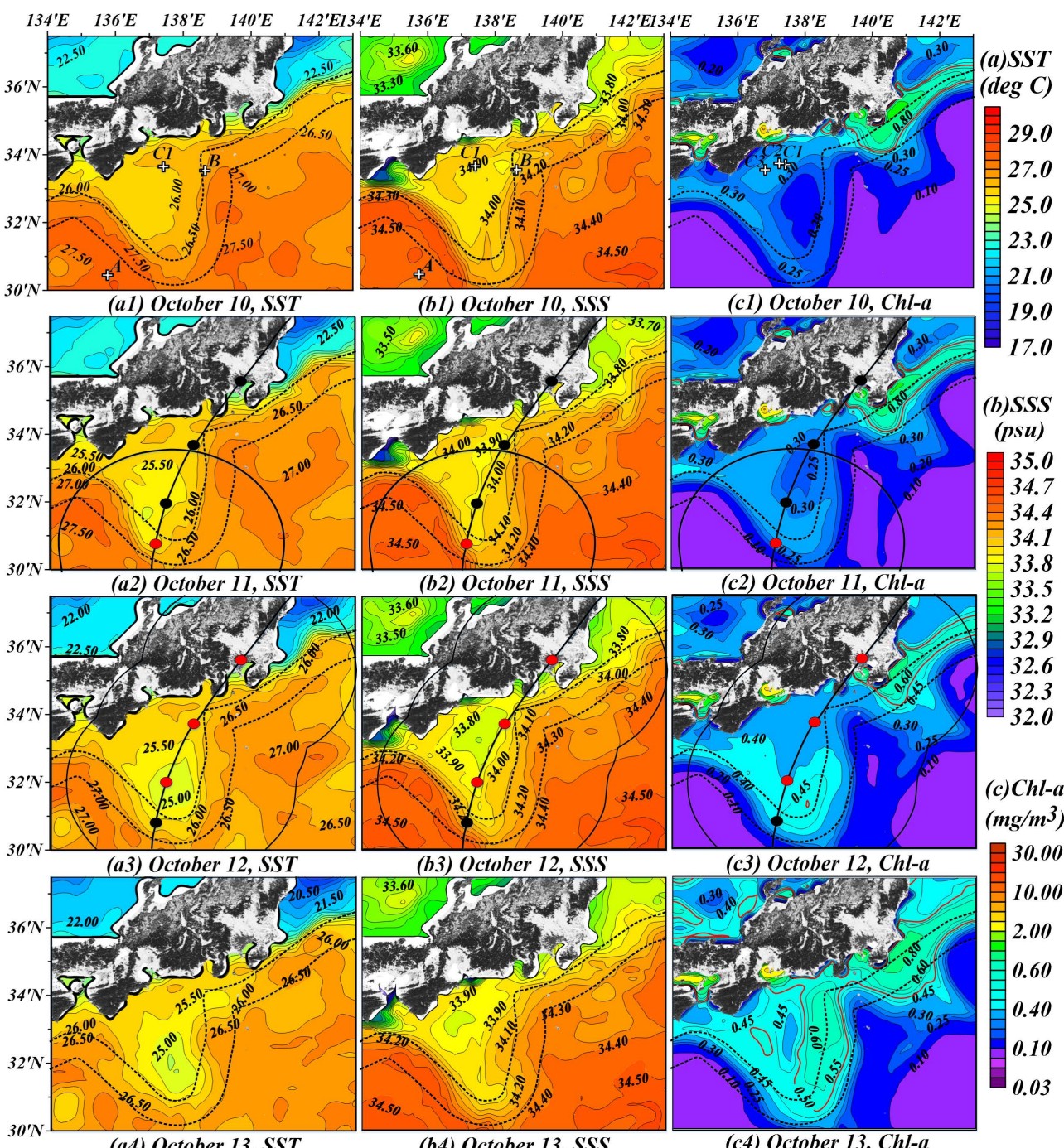

**Figure 8.** Daily sea surface temperature (SST), salinity (SSS), and chlorophyll a (Chl-a); on 10 October (**a1,b1,c1**), 11 October (**a2,b2,c2**), 12 October (**a3,b3,c3**), and 13 October (**a4,b4,c4**), respectively. Black dotted line, black line, and red dotted denote the Kuroshio meander provided by JCC, the storm wind zone, and the typhoon central location according to date. Solid red lines in all column c (**c1–c4**) indicate a boundary of high Chl-a is over 0.5 mg/m³. Physical Argo floats (ST and SS) are shown in points A and B, while a biogeochemical Argo float (ST, SS, DO, nitrate, and Chl-a) is shown in points C1, C2, and C3 according to date.

### 3.3.2. Cause Analysis through Spatial Distribution of SST, SSS, and Chl-a

Figure 8 shows the change in the SST, SSS, and Chl-a according to the passage of HAG-IBIS on 10–13 October 2019. The observed SST was 26.5 °C along the strong cyclonic eddy area formed above Kuroshio's path before the arrival of HAGIBIS in Figure 8(a1). When

HAGIBIS began affecting the study area, the SST changed from 26 to 25.5 °C at the strong cyclonic eddy (Figure 8(a2)). During HAGIBIS, it showed a temperature drop of about 0.5 to 1.5 °C inside SWZ in the strong cyclonic eddy area. It could be caused by the cold temperature rising from the deep water to the sea surface due to the combined upwelling effect of the typhoon and the cyclonic eddy. The magnitude of the SST cooling (more than 2 °C) depends not only on the intensity and translation speed of typhoons, but also on the preceding thermal structure (i.e., mixed layer depth and upper-ocean stratification) of the upper ocean. The thermal structure is addressed in the next section (Section 3.3.3. MLD and daily precipitation). One day after the typhoon, the decreasing SST remained at 25 °C in the strong cyclonic eddy zone (Figure 8(a4)).

The spatial distribution of SSS in Figure 8(b1) shows a low salinity concentration of at least 34.00 psu around the massive strong cyclonic eddy area before the arrival of HAGIBIS. SSS normally increases after a typhoon due to the vertical entrainment from wind-induced vertical mixing; however, Figure 8(b2) indicates a few changes in the SSS concentration, then a decrease of 0.2 as 33.80 psu on the left semicircle of the typhoon's center in the cyclonic eddy (Figure 8(b3)). The SSS remains with 33.90 psu in the same area until 13 October. A previous study [33] demonstrated that high salinity appeared on the eastern side of the typhoon track due to an asymmetrical effect as a dangerous semicircle. However, this study confirmed a massive decrease in salinity concentration on the left side of the typhoon track, whereas the SSS on the right side increased by a few concentrations and small extent. Compared to the effect of intense rainfall, the combined effect of strong Ekman and eddy upwelling may not greatly affect the increase in the SSS on the eastern and western sides of the typhoon's path. To interpret the causes of the change distribution of SSS, the following daily precipitation distribution will be analyzed in Section 3.3.3.

HAGIBIS had a wide SWZ encompassing the study area, strong wind stress power, and high Ekman pumping velocity. These characteristics significantly impacted biological processes in the ocean through direct or indirect effects on the PB. Before HAGIBIS, the PB existed along the KCM and in front of the offshore around the Kanto region, where the high Chl-a concentration remained at over 0.80 mg/m$^3$ within the red contour line (over 0.50 mg/m$^3$) due to the confluence area between the Oyashio cold and Kuroshio warm currents (location; longitude 140°–143°E and latitude 34°–36°N). This area gives rise to a natural fishing ground in Japan. In the cyclonic eddy area, the amount of Chl-a was from 0.25 to 0.30 mg/m$^3$ from 10 October to 11 in Figure 8(c1,c2). The Chl-a then reached 0.40 and 0.45 mg/m$^3$ along the Kuroshio path in the cyclonic eddy area (Figure 8(c3)). On 13 October, a massive growth of the PB (Chl-a reached 0.60 mg/m$^3$) occurred along the upper KCM and the center of the typhoon's path.

That is, HAGIBIS and accompanying heavy rainfall directly affected the decreasing SST and the low SSS on 12 October, while the large Chl-a anomalies occurred one day after the typhoon passed. These causes should be interpreted in conjunction with typhoon-related external and ocean internal factors simultaneously. This will be taken into account in the sea subsurface ocean section (Section 3.4).

### 3.3.3. MLD and Daily Precipitation

A relatively shallow MLD formed in a strong cyclonic eddy area, ranging from 30.0 to 35.0 m in thickness, was caused by pre-existing eddy upwelling as shown in Figure 9(a1). On 11 October, the sustained wind-induced energy flux triggered deepening MLD on the typhoon's center and right-hand side (occurring at more than 60 m depth; red line in Figure 9(a2)). On 12 October, the MLD in the cyclonic eddy area became shallow along the typhoon track center, with a value of 25.0 m depth in Figure 9(a3). This lifted-up MLD generated by the combined Ekman pumping and eddy upwelling could cause a decrease in SST, as shown in Figure 8(a3). In contrast, wind stress power deepens the MLD on the right and left sides of the typhoon path in the red contour area over 60 m to a maximum value of 83.6 m depth (Figure 9(a3)). The overall MLD remained in a similar condition or deepened on the northern side. In contrast, the MLD was thicker and the thermocline was

deeper within anticyclonic eddies (red area in Figure 9(a2–a4)). A thick mixed layer in the anticyclonic eddy was presumed to prevent deep, cold water from being entrained into the surface layer. Also, the warm water in the thick layer contributes to the intensification of typhoons and sustains the typhoon's intensity.

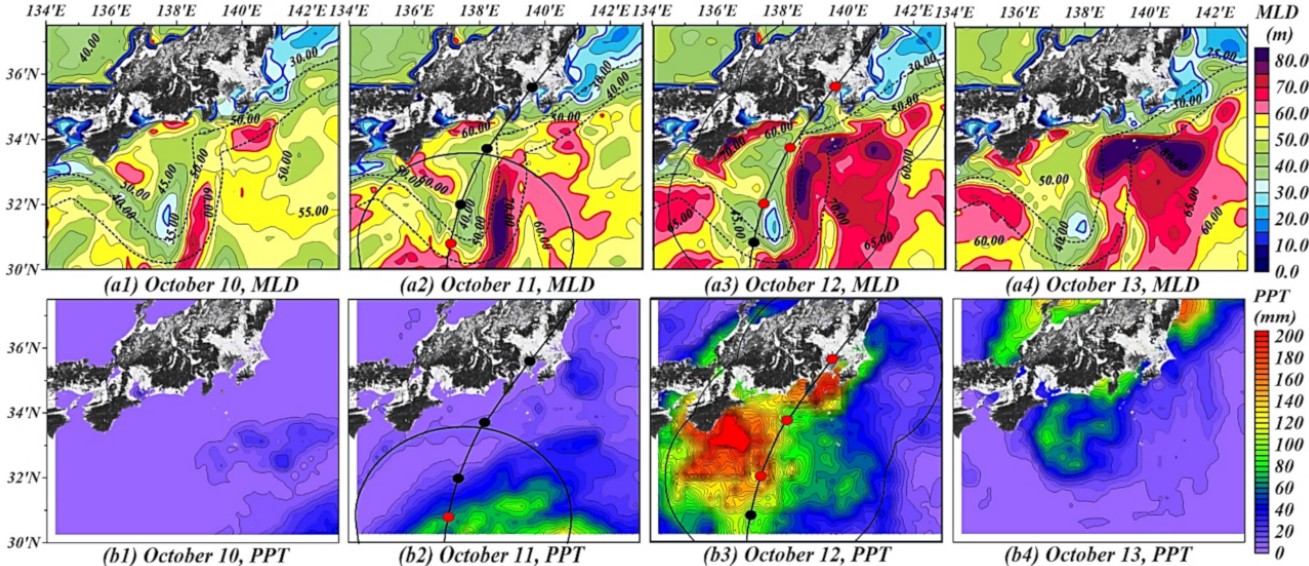

**Figure 9.** Daily mixed layer depth (MLD) and daily cumulative precipitation (PPT; mm); 10 October (**a1,b1**) 11 October (**a2,b2**) 12 October (**a3,b3**), and 13 October (**a4,b4**), respectively. Black dotted line, black line, and red dotted denote the Kuroshio meander, storm wind zone, and the location of the center typhoon corresponding date. The blue and red contour lines indicate 30 m and 60 m thickness in terms of MLD, respectively.

Figure 9b indicates the daily precipitation distribution observed by TRMM radar. Before the arrival of HAGIBIS in Japan, daily rainfall was approximately 100 mm. The typhoon on the left-bias sides was accompanied by heavy daily rainfall of over 200 mm. Due to the influence of rainfall, some oceanic variables changed. Particularly, the SSS decreased within the rainfall range during and after the typhoon's passage. Ekman pumping induced by HAGIBIS contributed substantially to the increase in the SSS; however, heavy rainfall effectively reduced SSS. This result is consistent with the conclusions of Sun et al. [31].

In summary, the ocean surface analysis revealed that the typhoon affected ocean values. The SST was reduced by the strong Ekman upwelling along the center of HAGIBIS's track in combination with the strong cyclonic eddy induced by the Kuroshio meander. The SSS was reduced by a massive supply of freshwater due to heavy rainfall on the left side of HAGIBIS's track, while the SSS increased slightly despite strong Ekman upwelling along the center of HAGIBIS's track. The MLD was deepened by strong wind stress power on the right and left semicircles of HAGIBIS, whereas the MLD shallowed along the center of the HAGIBIS's track due to the combined effect. However, to better understand the growth mechanism of PBs, we need to investigate why massive Chl-a occurs on the sea surface. Thus, the following section will elucidate the sea subsurface analysis.

### 3.4. Responses of Sea Subsurface Ocean Variables until 100 m Depth

3.4.1. Favorable Environmental Conditions in Phytoplankton Bloom

To investigate the biological process, we analyzed vertical changes in temperature, salinity, dissolved oxygen (DO), nitrate, and Chl-a as measured by a BGC Argo float at point C1; before (No.98 on 9 October), C2; one day after (No.99 on 14 October), and C3; a week after (No.100 on 19 October) the typhoon, as shown in Figure 8(c1). The location of point C1, C2, and C3 existed at the offshore sea of the Tokai region, where the eddy's strength was weak (−0.084 m) and the KCM's influence on the BGC Argo float was small.

On the contrary, HAGIBIS affected the area with strong Ekman pumping and heavy rainfall on 12 October. Therefore, we examined the direct impact on the center and left side of the typhoon rather than the combined effect, such as eddies, the Kuroshio, and the typhoon.

Temperature (°C, solid line), below the *x*-axis, and salinity (psu, dotted line), above the *x*-axis, are shown in Figure 10a. The temperature progressively decreased to 26 °C compared to before HAGIBIS (26.6 °C), and the thermocline depth was transformed by replying to the typhoon from 60 m to 40 m. The greater the water depth, the higher the temperature difference between before and one day after. Then, the ocean temperature showed a small variation of approximately 1 °C at around 100 m depth. This result is due to the Ekman upwelling of HAGIBIS bringing the deep cold temperature layer to the upper ocean. One week later, the sea temperature decreased to 25 °C near the surface, and the thermocline depth deepened to 50 m. The temperature at the bottom of the thermocline depth recovered to the levels before the typhoon. Meanwhile, based on the halocline on 9 October compared with 14 October, the salinity layer rose from 60 m to 40 m, similar to the thermocline in Figure 10a. A tendency for high SSS (34 to 34.2 psu) existed due to the prior influence of Ekman upwelling, and the heavy rainfall accompanied by HAGIBIS on 12 October stopped the supply of fresh water to the ocean. However, another heavy rainfall occurred on 19 October, and the total precipitation exceeded 200 mm in the NPO. As a result, SSS swiftly dropped to a relatively lower salinity of 33.6 psu after one week. The salinity change is due to the influence of the typhoon; however, the effect of rainfall was dominant in the vicinity of the surface layer until approximately 40 m deep.

Biogeochemical components such as DO and nitrate in Figure 10b show exchanges at each depth following the passage of HAGIBIS. The DO concentration from the surface to 30 m depth was comparable for each variable when compared with before and one day after the typhoon. However, at depths greater than 30 m, the mass changed considerably. It was reduced by 140 µmol/kg (before, 163 µmol/kg) at a 53 m depth due to typhoon-induced upwelling that could transport the low-oxygen in the deep water to the upper layer. It also showed relatively high oxygen at approximately 80 m depth compared with before and one day after HAGIBIS. Physical processes established the decrease in DO at 53 m depth. From another aspect, the DO result was inversely correlated with nitrate and Chl-a at a specific depth. The decreased (increased) DO on the specific layer is related to the consumption (release) of oxygen by organisms (primary production). Although there are differences in typhoons' intensity and ocean environmental conditions, the correlation is consistent with a previous result from Wang et al. [34]. A week later, the DO mass amount increased with oxycline depth, possibly due to the effect of the other heavy rainfall. The DO concentration returns to the initial condition below approximately 40 m depth. The nitrate concentration, the ocean nutrient index in the ocean, is monotonous despite the influence of HAGIBIS. The nitrate concentration largely increases due to strong Ekman upwellings bringing up a high amount of nitrate from deep water to the upper ocean one day after. One week after, the nitrate was restored to its original amount and depth compared with before HAGIBIS, as shown in Figure 10b.

Finally, the concentration of Chl-a in Figure 10c shows the nutrient-rich layer as 1 mg/m$^3$ (SCML; over 0.7 mg/m$^3$) remaining at a 60 m depth before the typhoon. One day after HAGIBIS, the SCML (0.75 mg/m$^3$) was pumped up to a 40 m depth, and then the SCML reached the sea surface, maintaining the high Chl-a concentration. A week after HAGIBIS, Chl-a concentration on the surface was sharply decreased up to 0.3 mg/m$^3$ whereas the redistribution of a nutrient-rich layer was 30 m. Considering the layer shift, the nutrient's favorable depth moved from 60 m to the upper ocean (less than 40 m deep), at which some ocean factors affect the PB's growth, such as the colder temperature, the lower salinity, and the higher oxygen regardless of the nitracline.

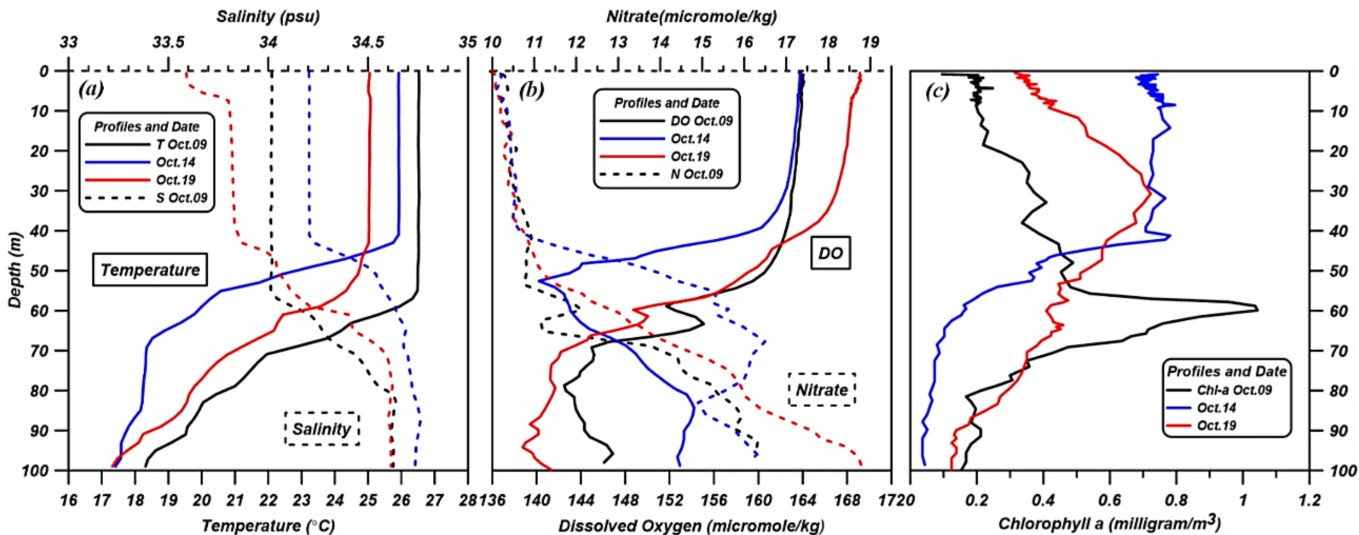

**Figure 10.** Depth-integrated (**a**) temperature (°C, solid) and salinity (psu, dotted), (**b**) dissolved oxygen (μmol/kg, solid) and nitrate (μmol/kg, dotted), and (**c**) Chl-a (mg/m³, solid) in the upper 100 m depth observed by Code 2,902,754 BCG Argo float before (C1; 9 October), a day after (C2; 14 October), and a week after (C3; 19 October) the passage of HAGIBIS, respecttively.

3.4.2. Comprehensive Impact Analysis

Until the preceding sections, individually estimated ocean responses to the typhoon's effects were analyzed and interpreted. For instance, on the sea surface, the typhoon mainly induced changes in the SST, SSS, and MLD due to the typhoon's physical impact. In contrast, the Chl-a occurred in the upper ocean layer one day after HAGIBIS. Therefore, further analysis is needed to reveal the causes using a comprehensive approach.

Figure 11 shows the depth-integrated ocean variability (before, during, and one day after), the KCM, and the typhoon effects simultaneously. Before HAGIBIS, the dominant horizontal distribution had a massive cyclonic eddy (the strongest area had over −1.0 m SLA) and a shift of Kuroshio's passage moving southeastward and northward with the current velocity over 1.0 m/s, propagating until 100 m depth (Figure 11a). The pre-existing cyclonic eddy induced eddy upwelling. Some ocean responses existed; firstly, the SST was decreased by the strong cyclonic eddy zone (Figure 11d), and the MLD was also shallower than in other areas (approximately 20.0 m depth at 138.3°E in Figure 11a). Interestingly, the Chl-a concentration on the surface was affected by the cyclonic eddy and the Kuroshio current area (136.3°–137°E and 138°E in Figure 11f). The PB occurred along the strong Kuroshio current and the cyclonic eddy area. These results agree with the discussion of Lizarbe et al. [35]. In addition, the high subsurface chlorophyll (HSC) layer existed around longitude 136.5°–137°E and 100 m to 80 m depth with 0.5 mg/m³.

During HAGIBIS, the effects of the typhoon area were distinct in each region. Firstly, HAGIBIS provided intensive stirring activity on the left (average over 30 W/m²; 135°–136.2°E) and right (over 35 W/m²; 138°–140°E) semicircles, and had relatively weak stirring in the center of the typhoon (25 W/m²) in Figure 11e. In contrast, the high EPV (136.2°–138°E) largely fluctuated from a minimum of 0.5 to a maximum of $50 \times 10^{-6}$ m/s (Figure 11e). Strong Ekman pumping led to a faster Kuroshio current velocity and a larger Kuroshio extent than before the typhoon. Interestingly, the area around the strong cyclonic eddy activity was also affected by the intense Ekman pumping. Thus, the combined effect induced changes in the MLD, SST, and Chl-a in the upper ocean layer. For example, some MLD became shallower at both high EPV (22.3 m depth) and intense Kuroshio current velocity (34.0 m depth at 136.5°E), while the other MLD in the vicinity of 135°E and 140°E became deeper than before the typhoon (54.2 and 63.2 m depth at anticyclonic eddy area in Figure 11b). Regarding temperature, the SST decreased until 24.6 °C at approximately

137.7°E, indicating the most intensive combined effect (Figure 11d). The HSC (0.5 mg/m³) was redistributed at a depth of 100 m to 45 m. Then, the HSC provided massive nutrients to an upper ocean layer to induce bloom. The surface Chl-a was approximately 0.5 mg/m³ (Figure 11f).

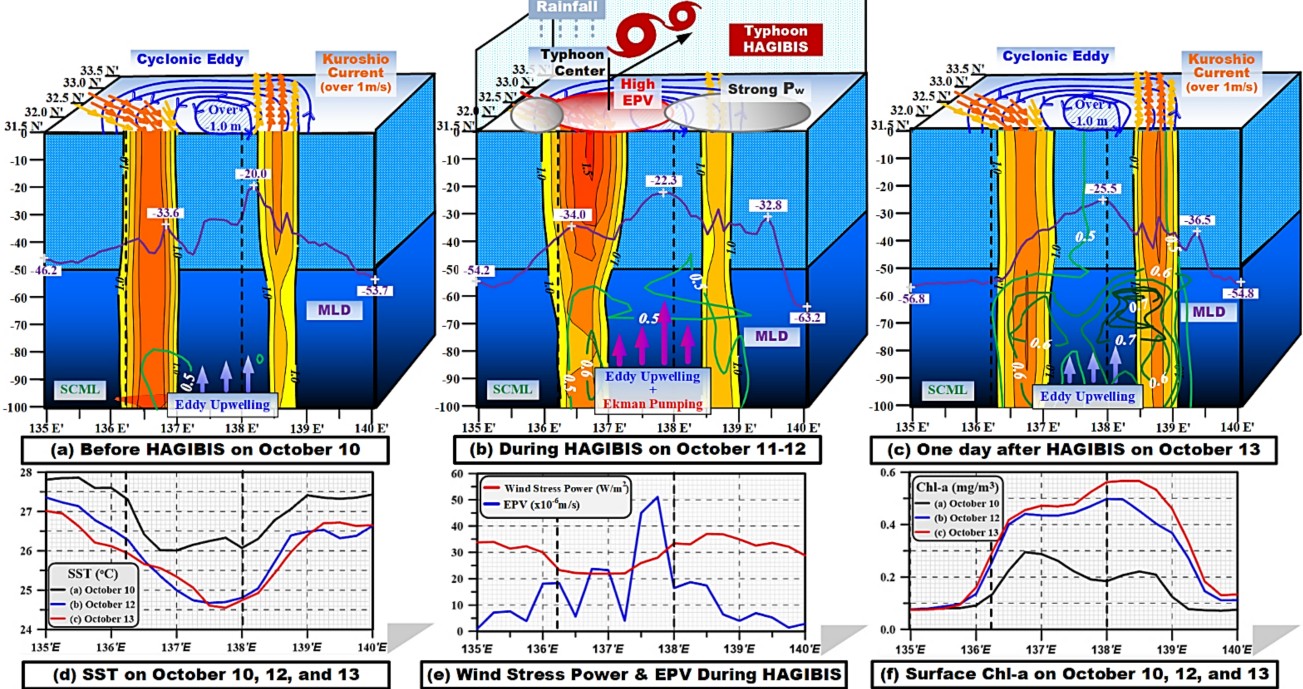

**Figure 11.** Quantitative conceptual diagram of ocean variability from 100 m depth to the sea surface; (**a**) before, (**b**) during, and (**c**) a day after HAGIBIS; Northwest Pacific Ocean zonal transect (latitude 31.5°N; longitude 135°–140°E) including cyclonic eddy area [blue isolines; (**a**–**c**) in the sea surface], the Kuroshio current depicted by integrated horizontal velocities on different depths with red; over 1.5 m/s, orange; 1.0 < V < 1.5 m/s, and yellow; over 1.0 m/s, and typhoon HAGIBIS represented by the affected area where strong $P_w$ (gray circles) and high EPV (red circle) at (**b**) in the sea surface. The subsurface layers comprise the MLD (Violet), HSC (light green; 0.5 mg/m³), and SCML (deep green; above 0.7 mg/m³). The three graphs indicate (**e**) $P_w$ (W/m²) and EPV ($\times 10^{-6}$ m/s) on the center panel during HAGIBIS, (**d**) sea surface temperature (°C) on the left panel, and (**f**) concentration of surface Chl-a (mg/m³) on the right panel before, during, and a day after. Two vertical dashed lines marked on both the main plot and the subplot indicate the region with the strongest typhoon effect along the typhoon's center (longitude 136.5°–138°E).

One day after HAGIBIS, the SST slightly decreased around the combined effect area (137.7°–138.2°E) and was intensive in the Kuroshio area (136.5°–137°E) (Figure 11d). The MLD also deepened in the same area. However, the fluctuation tendency of the SST and the MLD had approximately similar values during the typhoon, indicating the influence of the typhoon remained in the ocean, while the speed and range of the Kuroshio current returned to the pre-storm condition (Figure 11c). We previously showed the favorable environmental conditions in the PB's growth through the BGC Argo float at the specific locations C1, C2, and C3. The previous sections concluded that a modulated environment (decrease in temperature, low salinity, and high oxygen) caused by the typhoon effects could induce better conditions for PB growth. Furthermore, this comprehensive approach could be explained largely by the SCML (over 0.7 mg/m³) existing at 80 m to 60 m depth (138.0°–138.8°E), and the overall HSC (0.5 mg/m³) supplying the whole upper ocean layer from 100 m to 0 m depth (137.5°–139°E in Figure 11c). The nutrient-rich layers induced the massive PB at the sea surface one day after the typhoon. This explains why biological redistribution (HSC and SCML) is an important mechanism responsible for daily

surface PB. According to Wang et al. [34], they found different mechanisms causing surface phytoplankton blooms in the environment of the Arabian sea in response to successive tropical cyclones. However, the authors did not address the daily distinction between physical and phytoplankton dynamics. Our study demonstrated a significant difference in daily ocean variability for the typhoon period by employing a comprehensive approach to explain the daily variations in individually estimated physical and biological mechanisms.

## 4. Conclusions

In this study, the effects of the super typhoon HAGIBIS under the Kuroshio meander in the Northwest Pacific Ocean were investigated using the six-hourly wind product, daily ocean observational and model data for the sea surface, and Argo float and model data for the sea subsurface. For research question (Q1), the typhoon's effects were mainly wind stress power, which was stronger on the left and right semicircles than the center of the typhoon, while Ekman Pumping velocity was powerful in the center of the typhoon. The local characteristics of the typhoon are within storm wind zones. For research question (Q2), each typhoon effect induced a decreasing sea surface temperature, a shallowing mixed layer depth, and a high concentration of Chl-a. Notably, heavy rainfall reduced the sea surface salinity when the left side of the typhoon passed near the Japanese archipelago. However, the massive phytoplankton bloom on the sea surface occurred after one day. For research question (Q3), we conducted an expanded investigation for the favorable environment of the PB's growth and analyzed the sea subsurface to 100 m depth. The favorable depth shifted from 60 m to the upper ocean one day after the typhoon due to changes in ocean internal factors such as colder temperature, lower salinity, and higher oxygen regardless of the nitracline. The large and wide PB on the sea surface was caused by the redistribution of the SCML (above $0.7 \, \mathrm{mg/m^3}$), rising from a 100 m to 60 m depth and directly supplying high subsurface chlorophyll ($0.5 \, \mathrm{mg/m^3}$) to the sea surface. The nutrient-rich layer redistribution depended on the high Kuroshio current velocity area and the strong cyclonic eddy area. The comprehensive and quantitative impact analysis demonstrated the different spatial and temporal mechanisms between physical and biological oceanic variables in response to HAGIBIS. This may contribute to a better understanding of the ocean's physical and phytoplankton dynamics.

Furthermore, as the world begins to be severely affected by hazardous climatic events induced by global warming, this study is particularly relevant to present scenarios and of interest to the journal's readership. Our analysis highlights various data sources estimating ocean variability through their inhibition and growth using individually estimated analysis and comprehensive impact analysis. It may help in assessing ocean profiles in response to typhoons. In addition, our methodology may further support predicting the mapping area via a nutrient-rich zone. Future studies are needed to explore more specific events to determine the significance of our results for application to other study regions.

**Author Contributions:** J.J. and T.T. jointly conducted the research, processing, and analysis. J.J. worked on collecting data and preparing a quantification model and visualization, and writing a manuscript; T.T. designed, reviewed, and edited the research as supervision. All authors have read and agreed to the published version of the manuscript.

**Funding:** This research received no external funding.

**Acknowledgments:** We sincerely thank the "Typhoon HAGIBIS; Japan Meteorological Agency (JMA)" for offering data support. Thanks to "Wind product and Ocean Variables; Copernicus Marine Environment Monitoring Service (CMEMS)" for providing data support. Thanks to "Sea Level Anomaly; Arching, Validation, and Interpretation of Satellite Oceanographic (AVISO)" for offering data support. Thanks to "Rainfall; Goddard Earth Sciences Data and Information Services Center (GES DISC)" for supporting the data. Thanks to "Argo float data; Ministry of Science and Technology of China (MOST)" for offering data. We truly thank the five anonymous reviewers and the editor for their insightful comments and suggestions for improving this manuscript.

**Conflicts of Interest:** The authors declare no conflict of interest.

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
