# Peer review of "Investigating the Effects of Super Typhoon HAGIBIS in the Northwest Pacific Ocean Using Multiple Observational Data"

_remotesensing, doi:10.3390/rs14225667_

Round 1

Reviewer 1 Report

General commets

In this paper, the authors investigate the upper ocean response to the extreme event, typhoon HAGIBS, in the Northwest Pacific combined with the effect of the Kuroshio current meander. The work is very interesting, and the English is clear and easy to follow. The authors described the methods and results accurately. The work is fluent and pleasant to read. I certainly recommend the publication of the manuscript.

Specific Comments

Figure 8: define in the caption the points B and C depicted in subpanels (a), (b) and (c).

Lines 468-478: This paragraph is unclear, especially the reasons for choosing point C and the last sentence "Therefore, we examined the direct impact...." . In Figure 8a there is also a point B but it is not described in the text.

Figure 12: I suggest improving the caption accurately. For example: a) explain that the orange-colored area represents the vertical extent of the Kuroshio current, specify what the isolines in the Kuroshio current are (velocities? Are there vertical velocities? or are they horizontal velocities on different layers?); b) explain what the horizontal dashed lines are in all the subplots.

Figure 12c: Explain in the text why the SCML reaches the surface.

Reviewer 2 Report

Review on the paper "Investigating the effects of super typhoon
HAGIBIS in the 2 Northwest Pacific Ocean using multiple observational
data" by J. Jeon and T. Tomita

The paper offer an analysis of the combined effects of the typhoon
Hagibis and the Kuroshio current on the characteristics (Temperature,
Salinity, etc..) on the Northwest Pacific Ocean waters.  Data comes
from a variety of sources: from measurements (Satellites, Argo floats,
etc...) anf hybrid data (sparse observation complemented by modelling
such as CMEMS products).

The paper contains information that the readership of the journal, but
overall the paper suffers from default making it sometime unclear and
difficult to read.  English must be improved and the wording must be
carefully revised to offer precise, accurate information.

Please find below a non-exhaustive list of detailed point:

1) Overall, please check the use of the article "a/an", "the"
or their omission.  For example

l. 13 "a wind-induced mixing.."
l. 14 "an Ekman pumping velocity", etc...

2) Avoid ambiguous wording, for example

* The notion of "left" and "right" (e.g. l. 13 and elsewhere) are
inherently ambiguous until you have given a reference. Although they
are understood to be relative to the direction of motion of the
typhoon, this should be stated clearly in the text.

* l. 37 (and elsewhere) "oceanic variables" is too vague.  Specify
which variables or at least list the main ones if the exhaustive list
is too long.

* l. 48 (and elsewhere) I personally find the wording "typhoon's
effect" unclear.  Maybe "the influence of the typhoon" would clearer

3) l. 23-24: I do not understand the meaning of
"designing an efficient... area"

4) l. 83-84: the sentence "To better understand ...." is too vague
(again because "variables" is too vague).  The authors should be
more specific.

5) Captions of figure must be revised as not all symbols used are
defined in the captions.

6) Equation (2) is written in an ambiguous way.  "Curlz" is not a
standard notation.  The reader can only assume the authors mean the
vertical component of the vector curl(...).  In the argument of the
curl, the authors use a cross to denote a scalar multiplication which
is normally used for a vector product.

7) In subsection 2.3 Methogology, it would be clearer to use bullet
points for the various steps.

8) The full paragraph 3.1.1. lines 261-271 is unclear.

9) l. 336 An area cannot be "prevalent", please reword.

10)l. 338 "extended the area" to which area the authors refer to?

11) l. 339 "resonated" please explain.  Resonance has a specific meaning.

12) l. 402-403: Clarify here that the statement is supported by the
analysis below/ At first glance the statement seems unsubstantiated.

13) l. 469: Can the authors clarify whether the Argo float was
(relatively) stationary over the period of analysis?

14) l. 532... Define R^2 and p even if they are standard
notation/concepts.

15) l. 534.. The description of figure 11 in the text could be improved
by adding "The overall ocean temperature, salinity, DO and Nitrate
concentrations were were negligibly correlated with Chl-a as shown
by the blue markers and the blue line in Figure 11 ((a) R^2....." etc...

16) l. 551-552: The two sentence are ambiguous. Please rephrase.

17) Various wording suggestions

* Change all instances of "owing to" to "due to"
* l. 15. The authors use "Secondly" but have not used "Firstly" before.
* l. 31. Replace the semi-colon by a full stop and start a new sentence "The region experiences..."
* l. 36. References are needed after "... marine resources"
* l. 57 & 62. Why "mesoscale eddy" singular?
* l. 64. What is a KCM period?
* l. 97. "arrangeg" -> "organised"
* l. 99. "adopted method and methodology" seems to repeat roughly the same idea twice.
* l. 132. "Mulsourse" -> "Multi-source"
* l. 192. "disseminated" does not sound right in the context. Please reword.
* l. 237/8. meanders and eddies are features in the ocean rather than activities
* l. 260. "compared" -> "by comparison"
* l. 277. "expended" -> "expanded"
* l. 285. "oblique with". First "oblique" is an adjective, not a verb. Then "with" is not
correct in the context.  I assume the authors mean "when approaching Japan"
* l. 325. "could provide" -> "provides"
* l. 342. "progress" -> "motion"
* l. 345. insert "the one" between "similar to" and "before"
* l. 378. "transition of" -> "change in" (if this is what the authors mean).
* l. 474. "...influence was small with Argo floats"? Do the authors mean
         "... influence on the Argo floats was small"
* l. 484. "gap" Do the authors mean "variation"?
* l. 488. Insert "the levels" between "to" and "before"
* l. 499-501. "Compared with .... value" ->
  "The DO concentration .... value, compared to the one before HAGIBIS and the one
  one day after."
* l. 542. What is meant by "the salinity is altered". Do the authors mean
  "salinity decreases"?
* l. 562. "SST is cooled" -> "SST is decreased"
* l. 563. insert "in" between "than" and "another..."
* l. 639. "ocean science's physical..." -> "ocean's physical..."

Reviewer 3 Report

I think that this manuscript is well written. On the other hand, some discussion and/or explanation will be helpful for readers’ understanding. Considering them, I would like to accept this paper after moderate revision.

Comments:

- In Fig.5 & Fig.6, did you check SLP change during TC passage? In addition, how about TC’s translation speed? I think that Ekman upwelling does not occur enough if TC’s translation speed is too fast (faster than about 2m/s)..

- In Fig.11, did you derive the plots from point-by-point in each depths? If so, it may not contain physical meaning and thus may lead a confusion.. From Fig.10, Nutrient increase at subsurface (not surface) seems corresponding well with surface Chl-a increase.. I think that Fig.10 provides interesting information enough, while Fig.11 may be not so important (deleting Fig.11 may be better)..

Reviewer 4 Report

this paper is well written, apart from minor typos or expressions

it is well designed

more in-depth analyses could have been carried out but it is publishable

in present form

minor remarks

abstract

the following sentence sounds strnage

the combined effect of the 2019 super typhoon and of the KCM have led to ... changes since 2017

For Pw (everywhere) please do not write wind induced vertical mixing because this would depend on the ocean stratification and  vertical diffusivity; it is the wind stress power; it is the atmospheric mechanical energy reservoir for the vertical mixing in the ocean

line 332 We conducted THE STUDY OF the variability

line 500 was comparable FOR each variable

line 501 I do not understand what is meant by

"the mass was quite reformed deeper"

Reviewer 5 Report

This study provides us a comprehensive understanding of the effect of super typhoon in the NPO using multiple observational data. I agree with the acceptance of this ms. after a minor revision.

Some minor points could be considered as follows:

1.     The description “cooling sea temperature “might be not proper. “decreasing of sea temperature” maybe better. 

2.     Section 3.4.2, for R2 data, two valid numbers are generally used. For example, “R2=0.0496”, could be revised to “R2=0.05”.

Round 2

Reviewer 3 Report

I would like to accept in this present form.